# The Grain for Green Project in Contiguous Poverty-Stricken Regions of China: A Nature-Based Solution

**Tingyu Xu [1,2,3], Xiang Niu [1,2,3,*] and Bing Wang [1,2,3]**

1   Ecology and Nature Conservation Institute, Chinese Academy of Forestry, Beijing 100091, China; xty@caf.ac.cn (T.X.); wangbingcfern@163.com (B.W.)
2   Key Laboratory of Forest Ecology and Environment of National Forestry and Grassland Administration, Beijing 100091, China
3   Dagangshan National Key Field Observation and Research Station for Forest Ecosystem, Xinyu 338033, China
*   Correspondence: niuxiang@caf.ac.cn; Tel.: +86-10-6288-9561

**Abstract:** The Grain for Green Project (GGP) is one of many Nature-based Solutions (NbS), which aims to address the challenge of ecological restoration while providing livelihood security for farmers in poverty-dominated regions. Evaluating the success of such a project can prove difficult. Here, we choose the contiguous poverty-stricken regions (CPSR) of China to study the multiple benefits of the GGP in the context of NbS. We collect ecological-monitoring data, forest-resources data, and socioeconomic data and use them in a distributed method with relevant indicators, to evaluate the ecological benefits of the GGP. Additionally, the socioeconomic benefits are evaluated using questionnaire-based surveys. Our results showed that the ecological benefits of the GGP in the CPSR were $5.6 \times 10^{11}$ RMB/a in 2017, with the proportion of each ecosystem's services being 27.1% (water conservation), 21.1% (biodiversity conservation), 18.4% (purification of the atmospheric environment), 13.1% (soil conservation), 12.9% (carbon sequestration and oxygen release), 5.4% (forest protection), and 1.6% (nutrient accumulation). In terms of socioeconomic benefit, the GGP changed the production methods of farmers, which resulted in income growth, with an average increase of 5100 RMB/a per household. In the context of NbS, ecological conservation, and restoration, the accurate and systematic monitoring of the socioeconomic and ecological benefits will become more important for government decisions.

**Keywords:** Grain for Green Project; Nature-based Solutions; contiguous poverty-stricken regions; ecosystem services; socioeconomic benefits

## 1. Introduction

The contiguous poverty-stricken regions (CPSR) and related areas that need to implement special-support policies were delimited by the Chinese government in 2011 [1]. These areas are mainly distributed in the Tibetan Plateau, the Loess Plateau, the Inner Mongolian Plateau, the edge of the Gobi Desert, and the karst areas of South China [2]. Their remote locations, poor natural conditions, poor policy environment, and weak economic foundations as well as a lack of infrastructure, basic public services, and human capital are the major obstacles in targeted poverty alleviation [3].

Barron et al. [4] found that large, modern tomato-cultivation enterprises increase the income of migrants, which is fundamental to the survival of many villages in poverty-stricken regions. However, owing to a lack of capital investment and a lack of conditions that encourage indigenous local development, this industry cannot provide a complete solution to poverty. Jin et al. [5] found that there is a decoupling relationship between $CO_2$-emission reduction and poverty alleviation because of the contradiction between emission reductions based on limiting energy consumption and the alleviation of poverty. Hence, poverty alleviation in the region is not only an economic problem but also an ecological and cultural problem.

The contribution of ecosystems to deal with ecological and social challenges has been emphasized in many studies [6–8]. Nature-based Solutions (NbS) have been used in the context of managing aspects of climate change, biodiversity loss, water shortages, frequent natural disasters, human survival, and the sustainable development of society and the economy, which are all being severely damaged [9,10]. NbS encompass the practice of actively enhancing and protecting ecosystems as well as using ecosystem services to achieve sustainable-development goals, which transform the exploitation of nature from resource utilization to functional thinking, by making full use of ecosystem services to deal with the various severe challenges faced by human society [11]. NbS can support a wide range of sustainable development goals [12,13]. Eggermont divided NbS into three types, with the following aims [14]: (1) To make full use of natural or protected ecosystems with minimal interventions, by maintaining or improving their ecosystem-service functions. (2) To restore and manage the ecosystem, by taking the appropriate interventions to improve its ecosystem services. (3) To construct or build new ecosystems, with high-intensity interventions. Therefore, projects involving the restoration and rehabilitation of degraded ecosystems, such as the Grain for Green Project (GGP), can be seen in the context of NbS, to sustainably and successfully achieve sustainable-development goals.

The GGP, started in 1999, is mainly applied to areas with serious soil erosion and to cultivated land with a slope above 25 degrees, involving vegetation restoration and conversion to forest [15,16]. It is the largest in scale and the longest in duration of all the projects in China, also affecting the largest population and attracting the largest amount of investment [17,18]. The GGP involves 2435 counties (including county-level units) in 25 provinces across the country. Over the past 20 years, the Chinese central government has invested a total of RMB 517.4 billion into the project [19]. The project was conducted based on natural processes and cycles, as these use natural flows of matter and energy, which can improve the structure of natural ecosystems and then contribute to the society of a region as well as its sustainable economy, especially for fragile ecosystems that have been severely disturbed or destroyed by humans [20–24].

The ecosystem services produced by the GGP have included aspects such as a reduction in soil erosion [25] as well as improvements in carbon-storage and carbon-sequestration potential [26]. There have been many studies focused on evaluating the improvements brought about by the GGP [27] and on scientifically quantifying the benefits brought about by the implementation of the ecological engineering within the project [28]. Thus, studies aim to provide guidance for the further implementation of the project [29]. Additionally, some studies have found that the average household's real net income has increased significantly after participation in the GGP [30]. Hence, the GGP is not only a vegetation- and forest-restoration project but also a poverty-alleviation project. The project has also considered the social- and economic-development conditions of the project's targeted areas, as it has restored forest and grass vegetation (tailored to local conditions), to reduce natural disasters, optimize land-use structure, and increase the income of farmers [31].

However, in regards to the many types of NbS, there is a lack of systematic review regarding the assessment and quantification of the ecological and economic effects of the GGP and how it can solve social problems. Therefore, in this study, we attempt to quantify the ecological and socioeconomic benefits of the GGP in the CPSR. Further, as one of many NbS, we analyze how the multiple benefits generated by the GGP solve the associated challenges. A comprehensive evaluation of the multiple effects of the GGP in the CPSR is vital for strengthening the future implementation of the GGP in these regions. Thus, our results aim to provide a useful scientific basis towards improving policymaking, system planning, and project construction, to strengthen the GGP.

## 2. Materials and Methods

### 2.1. Overview of Grain for Green Project in Contiguous Poverty-Stricken Regions

The study area (Figure 1) was selected according to the regions defined by the Outline for Development-oriented Poverty Reduction for China's Rural Areas (2011–2020) [1], and

included 14 areas: the Dabie Mountains area, the Daxing'anling area, the four prefectures of southern Xinjiang, the Liupan Mountains area, the Luoxiao Mountains area, the Lüliang Mountains area, the Qinling–Bashan Mountains area, the Tibet Autonomous Region, the four provinces with Tibetan prefectures, the Wuling Mountains area, the Wumeng Mountains area, the Yanshan–Taihang area, the Yunnan–Guizhou–Guangxi rock-desert area, and the mountainous area on the western border of Yunnan.

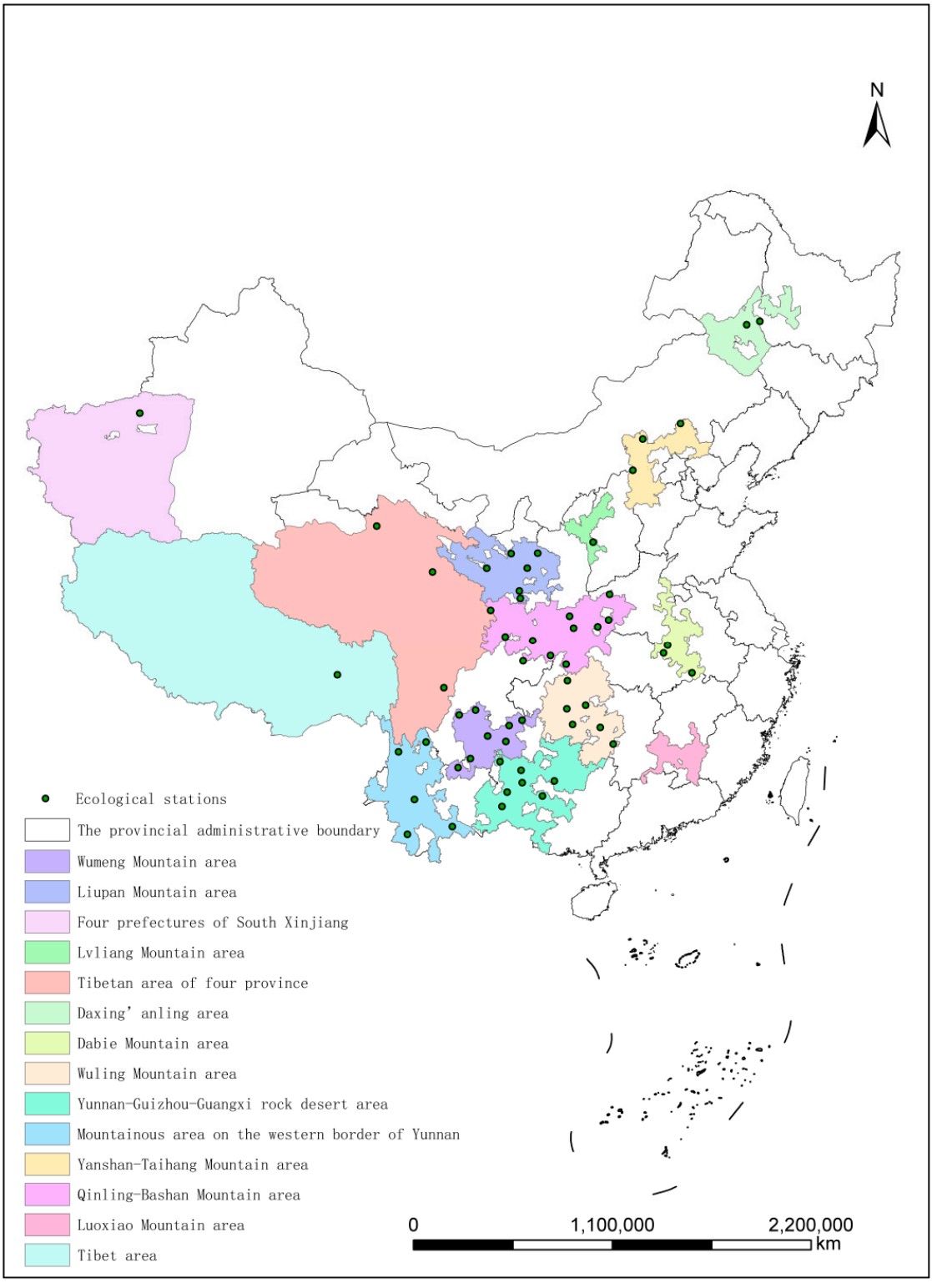

**Figure 1.** Distribution of monitoring sites in the CPSR of the study area.

By the end of 2017, the total area under GGP initiatives in the CPSR comprised $1256.94 \times 10^4$ hm$^2$. Various restoration methods (e.g., cropland afforestation, facilitate afforestation, and wasteland afforestation), inclusive of three forest types (e.g., ecological forest, economic forest, and shrubs), have been used in the ecological restoration of these areas (Table 1). Cropland afforestation returned farmland to forest, wasteland afforestation returned barren hills and wasteland to forest, and facilitate afforestation was achieved by closing forests to promote vegetation restoration.

**Table 1.** Areas of Grain for Green project (GGP) forest-restoration methods and forest types in the different contiguous poverty-stricken regions (CPSR) of the study area (Figure 1).

| CPSR | Total ($\times 10^4$ hm$^2$) | Restoration Methods ($\times 10^4$ hm$^2$) | | | Forest Types ($\times 10^4$ hm$^2$) | | |
|---|---|---|---|---|---|---|---|
| | | Cropland Afforestation | Wasteland Afforestation | Facilitate Afforestation | Ecological Forests | Economic Forests | Shrubs |
| Liupan Mountains area | 203.56 | 91.47 | 103.78 | 8.31 | 115.68 | 9.55 | 78.33 |
| Qinling–Bashan Mountains area | 223.55 | 101.26 | 104.79 | 17.50 | 160.01 | 56.30 | 7.24 |
| Wuling Mountains area | 169.86 | 80.18 | 78.20 | 11.48 | 142.04 | 25.11 | 2.71 |
| Wumeng Mountains area | 103.40 | 58.50 | 37.82 | 7.08 | 78.03 | 22.60 | 8.77 |
| Yunnan–Guizhou–Guangxi rock-desert area | 134.19 | 63.28 | 56.73 | 14.18 | 89.20 | 32.92 | 12.07 |
| Mountainous area on the western border of Yunnan | 81.17 | 36.05 | 36.80 | 8.32 | 51.35 | 24.79 | 5.03 |
| Daxing'anling area | 40.34 | 14.30 | 21.49 | 4.55 | 25.18 | 0.55 | 14.61 |
| Yanshan–Taihang Mountains area | 111.16 | 50.04 | 49.05 | 12.07 | 60.08 | 8.40 | 42.68 |
| Lüliang Mountains area | 65.06 | 26.18 | 35.77 | 3.11 | 35.30 | 9.32 | 20.44 |
| Dabie Mountains area | 36.40 | 9.97 | 23.40 | 3.03 | 29.26 | 6.33 | 0.81 |
| Luoxiao Mountains area | 19.80 | 4.66 | 11.74 | 3.40 | 18.25 | 1.42 | 0.13 |
| Tibet Autonomous Region | 3.65 | 2.53 | 0.61 | 0.51 | 2.40 | 0.32 | 0.93 |
| Four provinces with Tibetan prefectures | 30.69 | 16.46 | 9.42 | 4.81 | 20.32 | 3.74 | 6.63 |
| Four prefectures of southern Xinjiang | 34.11 | 17.56 | 13.37 | 3.18 | 12.69 | 11.37 | 10.05 |
| Total | 1256.94 | 572.44 | 582.97 | 101.53 | 839.79 | 212.72 | 204.43 |

### 2.2. Distribution of Monitoring Sites

GGP-monitoring sites were selected across the 14 CPSR (Figure 1). They include the 59 forest-ecological stations of the Chinese Forest Ecosystem Research Network (CFERN), 32 special stations for ecological monitoring of the GGP, more than 200 auxiliary monitoring sites for forest-ecological projects, and 7000 fixed sampling points.

### 2.3. Distributed Evaluation Method for Assessing Ecosystem Services

The GGP is a very large and complex project and is suitable to be divided into multiple homogeneous ecological-assessment units for evaluation [32–34]. We conducted a scale transformation of the ecosystem services, utilizing a distributed-evaluation method [34] (Figure 2). This method evaluated the ecological benefits associated with the GGP on five different levels, each at a smaller scale of detail, i.e., Level 1—CPSR (14); Level 2—admin-

istrative region (689); Level 3—restoration method (3); Level 4—forest type (3); and Level 5—dominant species. Ultimately, 35,828 relatively homogeneous assessment units were determined. Regional results were obtained by converting the results of all relatively homogeneous assessment units via forest-ecological correction coefficients and statistical principles [33].

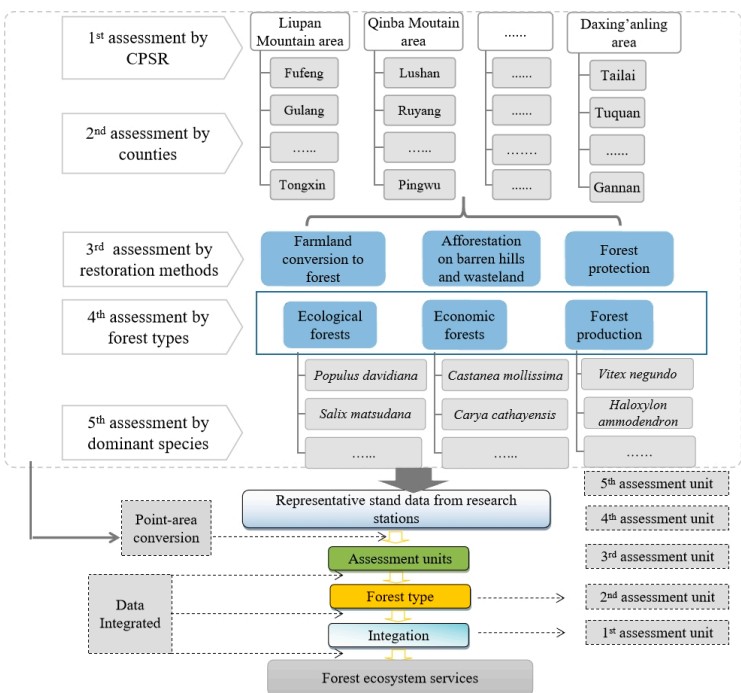

**Figure 2.** Schematic of the distributed evaluation method used in this study.

### 2.4. Data Collection for Assessing Ecosystem Services

Data used in this study were obtained by using long-term monitoring and conducting field surveys based on the national-standard system (Figure 1). Additional forest-resource data were provided by the State Forestry and Grass Administration of the GGP management center, and public data were collected from the relevant authorities. Socioeconomic data were collected from the counties within the GGP using surveys, questionnaires, and data from the provincial administrative departments. More details of the data collected are given in Figure 3.

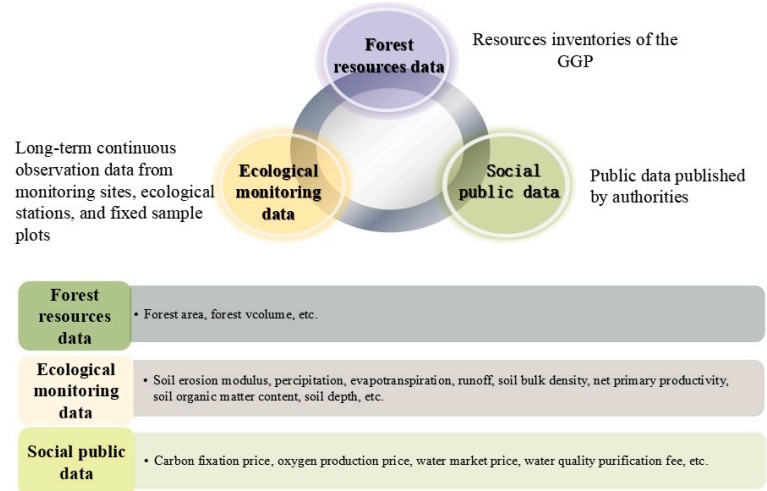

**Figure 3.** Summary of data collection used in this study for assessing ecosystem services.

### 2.5. Assessment of Ecosystem Serivices

Using the Specification of monitoring and evaluation of ecological benefits of returning farmland to forest project (LY/T 2573-2016) [35] and on the basis of representativeness, comprehensiveness, and actionability, we selected 7 categories of forest-ecosystem services: water conservation; soil conservation; carbon sequestration and oxygen release; nutrient accumulation; atmospheric environment purification; and forest protection and biodiversity conservation. When forest-ecosystem service quality was transformed into value quantity, it conformed to the principles of "equivalent substitution" and "weight-equivalent balance" [36]. We set 2017 as the base year of assessment.

### 2.6. Assessment of Socioeconomic Benefit

Social- and economic-benefit monitoring mainly focuses on the direct impact of the implementation of the GGP on the social and economic development in the CPSR, including the basic situation of the region's social economy; population resources and environment during the implementation of the project; the impact of the implementation of the project on rural-poverty alleviation, employment, the land-management system, production, and life; and regional economic development [37]. Table 2 gives more details.

**Table 2.** Monitoring indicators of socioeconomic benefits used in this study.

| Indicator | Details |
|---|---|
| Basic condition | Farmland area, forest area, grassland area, rural population, household situation, GDP |
| Social benefit | Poverty alleviation, employment by forestry, the development of emerging forestry-management entities, reform of the property-rights system, changes in production and lifestyle |
| Economic benefit | Forestry-output value, contribution to regional economic growth, regional industrial-structure adjustment, undergrowth economy, economic forests |

Data were obtained from a survey of social and economic benefits of the GGP conducted by the National Forestry and Grassland Administration using correspondence and interviews. There are 659 counties in the CPSR, which are also the main survey objects for the monitoring of socioeconomic benefits. The process was as follows: (1) In March 2018, a survey of the social and economic benefits of the GGP was carried out, conducted in 105 sample counties and 1576 sample households. (2) In January 2018, a questionnaire survey of farmers was performed using college students during their winter and summer vacations, covering about 8000 farmer households in 25 provinces. (3) In May 2018, a socioeconomic-benefit questionnaire was issued in the CPSR. (4) In January 2019, the survey data were reviewed and summarized in conjunction with the statistical bulletin. Data from the Monitoring of Social and Economic Benefits of National Forestry Key Projects from 2010 onwards were then added. Quality assurance of the monitoring and survey data was performed using data from the departments of statistics, agriculture, water conservancy, land, environmental protection, meteorology, and other similar departments.

### 3. Results

#### 3.1. Ecological Benefit of the GGP in the CPSR

The quantity of ecosystem services of the GGP in the CPSR is summarized in Table 3. The overall ecological benefit was $5.6 \times 10^{11}$ RMB/a, equivalent to 7.9% of the national forestry-output value, as well as equivalent to 6.5 times the total value of the central government's special poverty-alleviation program. Among the 14 areas, the Qinba Mountains area had the largest ecological benefit ($1.08 \times 10^{11}$ RMB/a), followed by the Wuling Mountains area, because of the large area of the project and the location of the area.

**Table 3.** Summary of the ecological benefits of the GGP in the CPSR.

| CPSR | Water Conservation (×10⁸ RMB/a) | Soil Conservation (×10⁸ RMB/a) | Carbon Sequestration and Oxygen Release (×10⁸ RMB/a) | Nutrient Accumulation (×10⁸ RMB/a) | Atmospheric Environment Purification (×10⁸ RMB/a) | Forest Protection (×10⁸ RMB/a) | Biodiversity Conservation (×10⁸ RMB/a) | Total (×10⁸ RMB/a) |
|---|---|---|---|---|---|---|---|---|
| Liupan Mountains area | 182.45 | 98.03 | 84.25 | 9.87 | 183.84 | 101.78 | 104.11 | 764.33 |
| Qinling–Bashan Mountains area | 333.64 | 118.76 | 146.66 | 21.11 | 219.37 | 48.20 | 195.63 | 1083.37 |
| Wuling Mountains area | 311.52 | 105.36 | 123.95 | 12.53 | 226.98 | —— | 188.83 | 969.17 |
| Wumeng Mountains area | 187.28 | 67.89 | 82.43 | 7.82 | 111.44 | —— | 121.47 | 578.33 |
| Yunnan–Guizhou–Guangxi rock-desert area | 170.38 | 49.29 | 88.71 | 6.53 | 135.90 | —— | 128.87 | 579.68 |
| Mountainous area on the western border of Yunnan | 168.32 | 53.77 | 56.40 | 3.62 | 71.98 | —— | 90.56 | 444.65 |
| Daxing'anling area | 37.32 | 13.49 | 19.92 | 2.04 | 32.11 | 13.96 | 15.63 | 134.47 |
| Yanshan–Taihang Mountains area | 66.09 | 26.17 | 90.49 | 2.22 | 55.57 | 21.80 | 23.17 | 285.51 |
| Lüliang Mountains area | 50.31 | 26.19 | 28.58 | 5.29 | 53.88 | 14.49 | 32.91 | 211.65 |
| Dabie Mountains area | 45.94 | 11.09 | 23.34 | 2.73 | 26.49 | 2.44 | 32.24 | 144.27 |
| Luoxiao Mountains area | 39.35 | 15.89 | 15.18 | 1.49 | 29.39 | —— | 19.91 | 121.21 |
| Tibet Autonomous Re-gion | 2.09 | 1.79 | 1.59 | 0.04 | 2.16 | 0.49 | 2.70 | 10.86 |
| Four provinces with Tibetan prefectures | 62.50 | 21.17 | 23.15 | 1.87 | 33.68 | 3.48 | 34.57 | 180.42 |
| Four prefectures of southern Xinjiang | 1.86 | 6.15 | 6.88 | 0.04 | 10.62 | 55.27 | 12.47 | 93.29 |
| Total | **1659.05** | **615.04** | **791.53** | **77.20** | **1193.41** | **261.91** | **1003.07** | **5601.21** |

The proportion of each ecological benefit value in the CPSR is shown in Figure 4. There are clearly notable differences between the areas. In most areas, water conservation and atmospheric purification are the main functions, and other indicators make relatively small contributions. The ecological benefits in the four prefectures of southern Xinjiang were mainly from forest protection, accounting for 59.2%. The ecological benefits in the Yanshan–Taihang Mountains area were mainly from carbon sequestration and oxygen release, accounting for 31.7%. There was no wind-proof or sand-fixing forest in the Wuling Mountains area, the Wumeng Mountains area, the Yunnan–Guizhou–Guangxi rock-desert area, the Luoxiao Mountains area, or the mountainous area on the western border of Yunnan, so their ecological benefit assessments do not include forest protection.

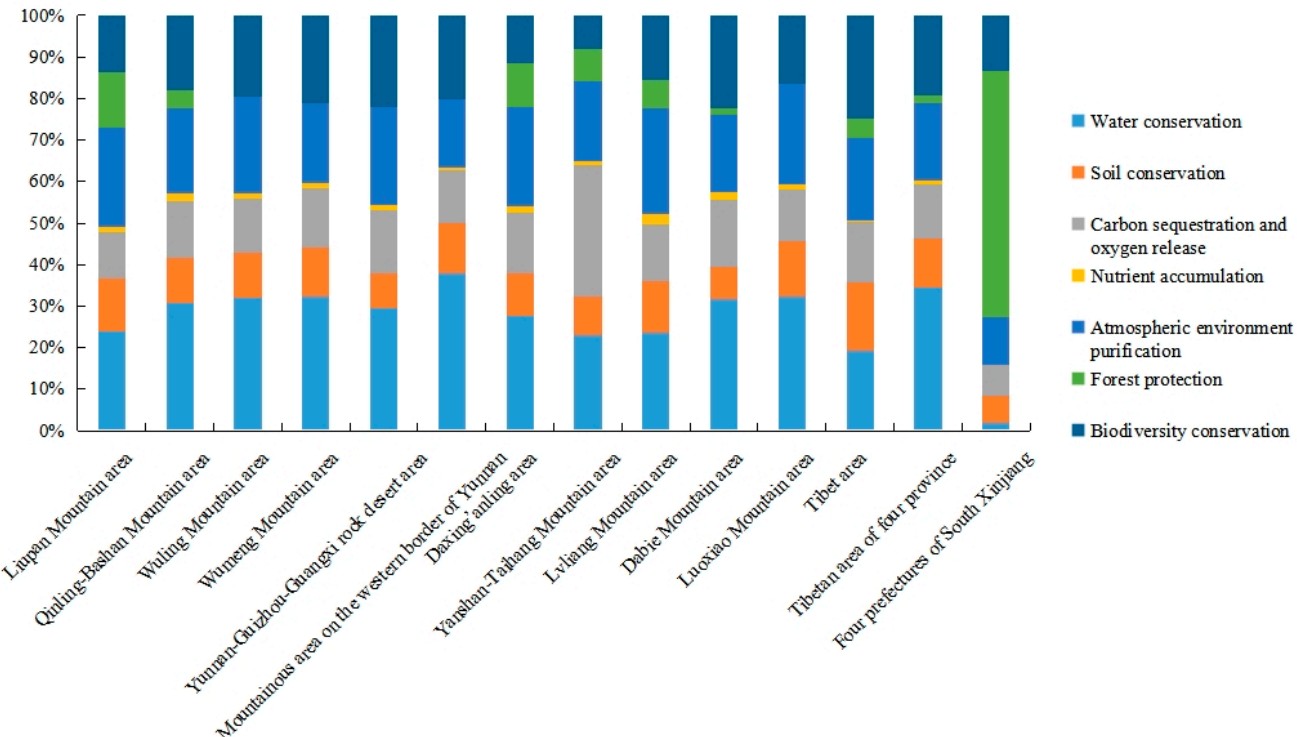

**Figure 4.** Proportion of each of the 7 categories of forest-ecosystem services in the 14 CPSRs of the study area.

### 3.2. Social Benefit of the GGP in the CPSR

The implementation of the GGP can improve the rural environment and help promote the rural-revitalization strategy [38]. In 2017, 66% of the afforestation tasks were implemented in the CPSR, and 399 townships and 98,600 households in these areas participated in the project. In terms of the poverty alleviation associated with the GGP, there were 2.2 million filed poor households in the monitored counties that participated in the project, accounting for 31.3% of households in the CPSR (Table 4). The proportions of the four provinces with Tibetan prefectures and the Wumeng Mountains area were larger than other areas. The monitoring results showed that nearly 60% of the woodlands within the GGP area had become forest, and 90% of the trees were growing well.

The forestry-employment rate of farmers on reclaimed farmland was 8.0%. Home interviews confirmed that 76.6% of the interviewed farmers manage their own land, and 23.4% of the interviewed farmers adopt cooperative-management methods or transfer their labor to large households. Migrant labor and forest management had promoted the employment of farmers. A large amount of labor was released from the primary industry and invested in secondary and tertiary industries.

**Table 4.** The number of filed poor households participating in the GGP and percentage of the total number in each CPSR.

| CPSR | Households/(×10⁴) | Percentage of the Area/% |
|---|---|---|
| Liupan Mountains area | 2.50 | 23.08 |
| Qinling–Bashan Mountains area | 43.75 | 38.89 |
| Wuling Mountains area | 62.41 | 38.52 |
| Liupan Mountains area | 42.84 | 53.06 |
| Wumeng Mountains area | 23.89 | 24.55 |
| Yunnan–Guizhou–Guangxi rock-desert area | 11.50 | 30.97 |
| Mountainous area on the western border of Yunnan | 0.09 | 16.27 |
| Daxing'anling area | 2.50 | 23.08 |
| Yanshan–Taihang Mountains area | 13.45 | 35.99 |
| Lüliang Mountains area | 5.10 | 26.48 |
| Dabie Mountains area | 2.53 | 7.34 |
| Luoxiao Mountains area | 2.50 | 3.91 |
| Tibet Autonomous Region | 0.98 | 35.01 |
| Four provinces with Tibetan prefectures | 3.91 | 70.86 |
| Four prefectures of southern Xinjiang | 1.18 | 4.04 |

### 3.3. Economic Benefit of the GGP in the CPSR

In 2017, the GDP of the monitoring counties was RMB $4.49 \times 10^{12}$, and the contribution rate of the forestry industry was 3.3%. By region, the forestry-industry-output values in the Wuling Mountains area, the mountainous area on the western border of Yunnan, the Yunnan–Guizhou–Guangxi rock-desert area, the Qinling–Bashan Mountains area, the Yanshan–Taihang Mountains area, and the Dabie Mountains area were relatively high, reaching more than RMB 10 billion in each area (Figure 5). After the implementation of the GGP, the structure of the regional-output value has been gradually optimized, the proportion of the output value of the primary industry has gradually declined, and the proportion of the output value of the tertiary industry has risen sharply. In 2017, the proportions of three industries were 6.8%, 20.5%, and 72.8%, respectively.

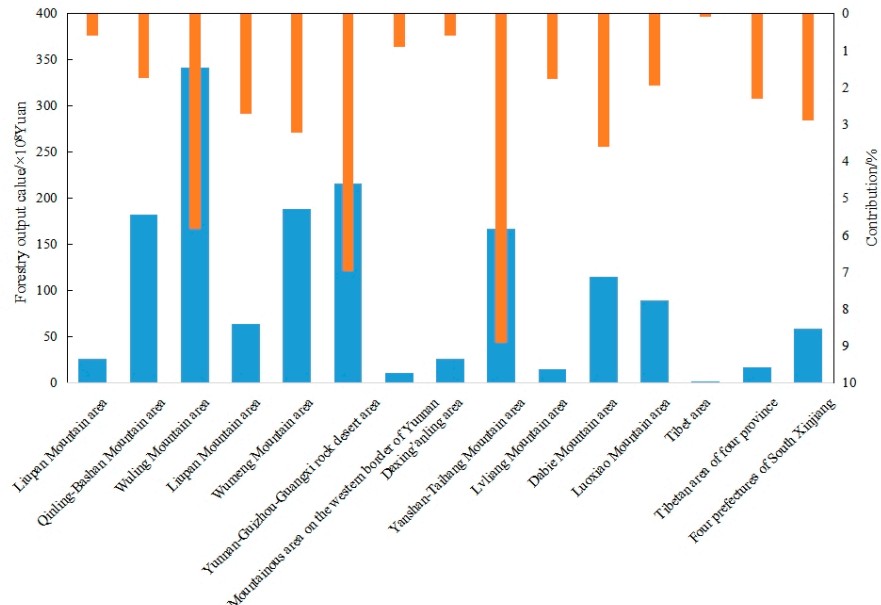

**Figure 5.** Forestry-output value and the percentage of forestry-output value compared to regional GDP in the CPSR.

Additionally, the understory economy (carried out under the canopy, such as agriculture, animal husbandry, herbal medicine, and medicine) developed quickly, and the production of fruit, herbs, and other economic-forest products increased significantly (Table 5). After the implementation of the GGP, the income from farmers' forestry productions and operations had increased significantly. What's more, multiple income channels, such as income from forestry-related work, property income, and financial aid, helped to increase household income. Compared with those who did not participate in the project, the income per capita increased by 1.5 times in 5 years.

**Table 5.** Forest-industry development in CPSR in 2017.

| Product Type | Yield | Percentage of Total Forestry Production/% |
|---|---|---|
| Bamboo/($\times 10^8$ roots) | 1.8 | 0.3 |
| Herb/($\times 10^4$ tons) | 34.4 | 12.3 |
| Fruit/($\times 10^4$ tons) | 225.2 | 3.5 |
| Understory farming/(RMB $\times 10^8$) | 690.1 | 27.4 |
| Understory planting/(RMB $\times 10^8$) | 434.3 | 35.3 |

## 4. Discussion

### 4.1. Contribution of the GGP in the Context of NBS in the CPSR

The standard aim of NbS is to solve multiple challenges through ecosystem services, including addressing climate change, improving biodiversity, building a sustainable food system, responding to the water crises, mitigating natural disasters, protecting human health, and promoting economic development sustainably [10,13], after the implementation of the GGP, which has been also reported by other researchers [39–41]. In total, four dominant functional groups of the actions within the GGP can be identified: a "green water pool" corresponding to the challenge of water security and disaster-risk reduction; a "green carbon pool" corresponding to the challenge of climate change mitigation and adaption; a "purify the environment and oxygen bar" corresponding to the challenge of human health; and a "biodiversity gene pool" corresponding to the challenge of environmental degradation and biodiversity loss [8]. These functions are vital to cope with the multiple challenges faced in the context of climate change. The key characteristic in the context of NbS is their capability to simultaneously perform multiple functions to deliver a set of associated ecosystem services. However, in this study, we did not consider cultural-ecosystem services [42], such as ecological tourism and ecological healthcare, which may be an important means for farmers to improve their living standards.

The GGP in the CPSR brought social and economic benefits. It promoted targeted poverty alleviation, employment of farmers, the development of new forestry-management entities, and improved rural equity as well as changed the production and lifestyles of farmers. The project promoted the transfer of rural labor. Previous studies have found that the implementation of the GGP has generated a large number of surplus laborers [28], who are switching from farming to non-farm work, which can provide higher earnings than crop production [43–45]. It can also stimulate the vitality of regional economic development and can alleviate poverty through financial subsidy, forestry production, and increased productivity of farmland [46].

The GGP is a "three-effect unification" of ecological, economic, and social benefits, and a systematic project of the "three combinations" of rural production and life and ecological improvement, which can provide the "win-win" gain of restoring the environment and promoting socioeconomic development [47]. The project can accelerate rural revitalization and attractive rural construction as well as improve regional-developmental motivation and capacity for farmers. Using forest employment to increase income provides new opportunities for rural-poverty alleviation and a new focus on promoted regional development and provided new economic-growth points [48].

### 4.2. Main Problems in the Process of Project Implementation

The first problem was that some farmland had to be returned to forest. We found that in the sample counties in 2017 (Figure 6), there were 1.6 million hectares of cultivated land with a slope of 25 degrees, facing severe desertification and rocky desertification, cultivated land with a slope of 15–25 degrees and important water sources, and severely polluted cultivated land. These degraded areas accounted for 26.7% of the total cultivated-land area. Basic farmland among the returnable farmland accounted for 12.9% of the total farmland area of the sample counties.

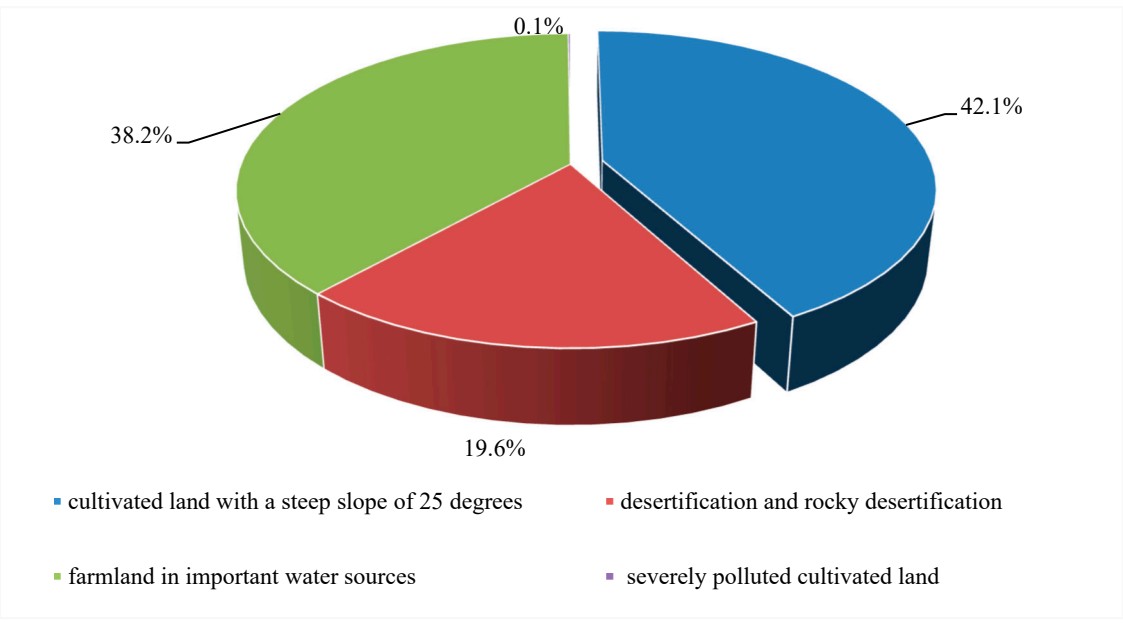

**Figure 6.** Different types of farmland with ecological problems.

We found that the main reasons why cultivated land should be returned or not returned to forest in some areas were as follows. (1) There were specific policies in some provinces concerning the amount of cultivated land with a steep slope of 25 degrees, facing severe desertification and rocky desertification, and cultivated land with a slope of 15–25 degrees and important water sources, which are not suitable for cultivation. According to the Regulations on the Protection of Basic Farmland [49], it is difficult for these sloping-farmland areas to be included in the scope of returned farmland. (2) The implementation of basic farmland-reduction policy is not in place. The lack of coordination between farmland protection and the policy of returning farmland to forest has seriously hindered the smooth progress of the new round of the GGP. (3) The Chinese government clearly stipulates that the identification of non-basic farmland on steep-slope cultivated land needs to use the current land-use and planning maps, and the quality of these maps can be variable. Only the combination of both types of maps can determine the land area to be returned to forest. (4) The integration of the new round of returning farmland to forest with grassland requires the data from three national land surveys. If the returnable farmland shown on the map cannot be found in situ, or the mapped area does not match reality, the task is then difficult to implement.

The problem of the fragmentation of abandoned farmland is prominent, which is not conducive to a large-scale operation. Due to the sporadic distribution of returned farmland, comprehensive management cannot be carried out in some watersheds and regions, which not only fails to achieve the original intention of the GGP for ecological restoration but also restricts the development of large-scale industries, which in turn affects farmers' enthusiasm for increasing their income and increasing regional economic and social-development speed. In addition, in some areas with harsh ecological environments,

the area of farmland returned to forest is often large, involving a large number of farmers involved in the GGP who often deeply impoverished. Economic forests in such areas often face difficulty to survive, so only ecological forests can be planted. The environment restricts the development of the under-forest economy, and it is difficult to develop follow-up industries upon returning farmland to forest, which has an impact on the livelihood of farmers.

### 4.3. Countermeasures and Suggestions

Li [28] found that the ecosystem services and economic benefits of the GGP have a high degree of coordination, as 90% of regions achieved extreme coordination and coupling. This means that when the ecosystem-service function of the GGP is improved, the CPSR achieve a high level of economic benefits. This confirms that "green water and green mountains" is a necessary developmental underpinning for "golden water and silver mountains" [28].

Forestry has great potential in poverty alleviation, by improving the subsidy policy for returning farmland to forest and grassland, appropriately increasing the subsidy standard, and extending the subsidy period to help poverty alleviation and consolidate the achievements of ecological construction. To achieve this, we put forward the following recommendations. (1) Expand the scale of the new round of returning farmland to forest and grassland as soon as possible, check that the land that can be returned based on the data of the three national land surveys, and select some of the abandoned farmland with large areas and good land conditions. (2) Establish a long-term mechanism to consolidate achievements, include expired farmland in the central and local forest-ecological-benefit compensation funds, and make a stable and long-term subsidy policy for returning farmland to forest. Include returning farmland to forest land into the local policy-based forestry-insurance system, innovate subsidy standards, optimize forest-land-contract management policies, and encourage and guide enterprises, large households and other industrial and commercial capital to reasonably transfer afforestation land. (3) The characteristic economic-forest industry, forest tourism industry, and health-care industry are the starting points to promote the implementation of the project of returning farmland to forest with economic benefits, strengthen industrial development as well as scientific and technological support, and improve the quality of forests. 4. Enhance the self-development ability of farmers who have returned farmland without destroying the vegetation and causing soil erosion; farmers can intercrop short crops such as beans, develop the under-forest economy, adjust the agricultural industry structure with the central government's special funds, and develop characteristic industries. Increase the income of households returning farmland, and consolidate the achievements of returning farmland to forest and grassland.

### 5. Conclusions

In the context of Nature-based Solutions, the Grain for Green Project produced multiple benefits in the contiguous poverty-stricken regions and will help realize the sustainable-development goals of China. The Grain for Green Project in the contiguous poverty-stricken regions has achieved the "win-win" gains of restoring the environment and promoting socioeconomic development. The ecological benefit of the Grain for Green Project in the contiguous poverty-stricken regions was $5601.21 \times 10^8$ RMB/a in 2017, with the proportion of each function being 27.1% (water conservation), 21.1% (biodiversity conservation), 18.4% (purification of atmospheric environment), 13.1% (soil conservation), 12.9% (carbon sequestration and oxygen release), 5.39% (forest protection), and 1.6% (nutrient accumulation). Among the 14 areas of the project, the Qinba Mountains area had the largest ecological benefit, accounting for 19.3% of the total, followed by the Wuling Mountains area, the Liupan Mountains area, the Yunnan–Guizhou–Guangxi rock-desert area, and the Wumeng Mountains area. In the contiguous poverty-stricken regions, 31.3% of the recorded poor households participated in the GGP, and the project changed the production methods of farmers, the forestry industry was promoted, and the income of farmers was increased,

with an average increase of $0.51 \times 10^4$ RMB/a per household. Further research should focus on interaction effects and measures to improve the socioeconomic–ecological benefits.

**Author Contributions:** B.W. conceived and designed the paper. X.N. analyzed the data and revised the manuscript. T.X. wrote the paper. All authors have read and agreed to the published version of the manuscript.

**Funding:** This study was funded by the Central Non-profit Research Institution of CAF (CAFYBB2020ZD002-2), Central Non-profit Research Institution of CAF (CAFYBB2020ZE003). Financial support was also provided by CFERN & Beijing Techno Solutions Award Funds for excellent academic achievements.

**Conflicts of Interest:** The authors declare no conflict of interest.

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
