# Peer review of "The Grain for Green Project in Contiguous Poverty-Stricken Regions of China: A Nature-Based Solution"

_sustainability, doi:10.3390/su14137755_

Round 1
Reviewer 1 Report
This paper aims to estimate the ecological and social-economic benefits of the "Grain for Green Project" in contiguous poverty-stricken areas. The topic is of great significance. It can be seen that the authors have also worked carefully in data collection and data processing. However, the article has the following problems, which are open for discussion.
- The research content seems not entirely consistent with the research question. First, concerning ecological benefits, this paper used the ecosystem services data in 2017. However, this data can only measure the stock of the ecological benefits of natural resources, and cannot be used to measure the benefit increment brought by the "Grain for Green Project". Second, in estimating the social-economic benefits, this paper used the survey data collected in 2018, and estimated the social-economic benefits based on a direct comparison between groups, without controlling other factors. As the two compared groups may be different in many factors, their difference could not be claimed as the causal effect of the Grain for Green Project".
- The validity of variables and samples is not addressed. Are the variables selected in this paper reasonable? Can they be used to measure forest resources' social-economic benefits? The paper does not address these issues. Furthermore, what is the validity of the survey data? Are the samples representative? There is no explanation.
- There are also many problems in writing. For example, some pictures and tables do not correspond to the content of the article. Section 3.1 mentions the value of "carbon sequestration and oxygen release". However, neither Figure 4 nor Table 3 have relevant information.
Author Response
Response to the Review Comments
We would like to thank you for your careful reading, helpful comments, and constructive suggestions, which has significantly improved the presentation of our manuscript.
We have carefully considered all comments from the reviewers and revised our manuscript accordingly. The manuscript has also been double-checked, and the typos and grammar errors we found have been corrected. In the following section, we summarize our responses to each comment from the reviewers. We believe that our responses have well addressed all concerns from the reviewers. We hope our revised manuscript can be accepted for publication.
- Thank you for pointing out this problem in manuscript.In this study, 2017 is the base year for the assessment. This paper calculates the inter-annual situation of ecological service functions since the implementation of the project of Grain for Green project in contiguous poverty-stricken regions, which represents the annual ecological benefits on the base year of 2017 , belonging to ecosystem service flow.The socio-economic data collected in 2018 which reflected the situation in 2017 or before, so the ecological and socio-economic benefits can be compared, and the farmers participating in GGP had changed the employee mode and increased the income also. These indicators are also used in “Monitoring of Social and Economic Benefits of National Forestry Key Projects” organized by National Forestry and Grassland Administration to reflect the changes in the living conditions of farmers after the implementation of the project.
- The Economic Development Research Center of the State Forestry and Grassland Administration has been carrying out the "Monitoring of the Social and Economic Benefit of National Forestry Key Projects" for a long time. The State Forestry and Grassland Administration's Project Management Office of Grain for Green project entrusted the center to design survey indicators, and review, summarize and analyze survey data. At the household level, the center commissioned college students from Beijing Forestry University and other forestry research institutes to conduct household interviews to ensure the independence and impartiality of the survey. In order to ensure the quality of the monitoring and survey data, the method of review and sampling was adopted, and the data of the departments of statistics, agriculture, water conservancy, land, environmental protection, meteorology and other departments at the same level were used to verify the survey data to ensure that the survey data were not false, concealed or wrong. We have added this section in the manuscript on line 173-188.
- We have double checked the chart and the corresponding description to make sure there were no errors.
Reviewer 2 Report
- The abstract part should highlight the innovation points and marginal contribution, and suggest the author to make relevant modifications.
- Introduction part does not describe the choice concentrated the contiguous poverty-stricken regions as the cause of the research object, the article only caused poverty in contiguous areas objectively, and realize is not only an economic problem, but ecological and cultural, so what the contiguous poverty-stricken regions have ecological and cultural value is worth studying? Some research papers have been well described, and the author is suggested to expand and analyze on this basis.
Is there a decoupling relationship between CO2 emission reduction and poverty alleviation in China? Technological Forecasting & Social Change. 2020.
The impact of the tomato agroindustry on the rural poor in Mexico. Agricultural Economics. 2000.
- Cropland Afforestation repeats in the name of restoration table in Table 1. The data reference of these two columns is unclear and should refer to cropland afforestation and wasteland afforestation in the original text
- The titles of 2.1 and 2.2 should be logically juxtaposed. Why the same titles? The same problem exists 2.4 and 2.5
- The new proper noun in title 2.3 is not scientific, for example: homogeneous ecological assessment units
- Figure 2 is the core of the distributed evaluation method in this paper. What is the operation logic of the evaluation method in Fig? Figure 2: How to transform the ecosystem service scale? What is the scientific basis for the division of one, two, three, four and five levels? What is their logical relationship? It is suggested to supplement and improve.
- In title2.6, What is the scientific basis for the selection of the indicators of the social and economic benefit evaluation system? Are the systems constructed by the other studies applicable in this paper?
- The Conclusion part is a little thin, as recommended.
Author Response
Response to the Review Comments
We would like to thank you for your careful reading, helpful comments, and constructive suggestions, which has significantly improved the presentation of our manuscript.
We have carefully considered all comments from the reviewers and revised our manuscript accordingly. The manuscript has also been double-checked, and the typos and grammar errors we found have been corrected. In the following section, we summarize our responses to each comment from the reviewers. We believe that our responses have well addressed all concerns from the reviewers. We hope our revised manuscript can be accepted for publication.
- Thank you for pointing out this problem in manuscript.We have adjusted and revised the sentence of the abstract to highlight the research content and innovation of the article.
The Grain for Green Project(GGP) is a NbS which aims to address the challenge of ecological restoration while providing providing livelihood security to farmers in a poverty dominated region. Evaluating the success of such a project can prove difficult. Here, we choose the contiguous poverty-stricken regions(CPSR) to study the multiple benefits of GGP as a NbS.
- Thank you for the suggestion.We added the following.
Barron et al.[4] found that the large, modern tomato cultivation enterprises increase the migrant income which is fundamental for the survival of many villages in the poverty-stricken regions. However, owing to a lack of capital investment and a lack of the conditions that encourage endogenous local development, this industry cannot provide a complete solution to poverty alleviation. Jin et al. found that there is a decoupling relationship between CO2 emission reduction and poverty alleviation because of the contradiction between emission reductions based on limiting energy consumption and the alleviation of poverty. Hence poverty alleviation in the region is not only an economic problem, but also an ecological and cultural problem.
- Thank you for pointing out this problem. It should be wasteland afforestationand we have corrected it.
- Thank you for pointing out this problem. It should be
2.2 Distribution of Monitoring Sites
2.5. Assessment of Ecosystem Serivices
We have corrected it.
- This term has been published in
Wei, W.J.; Wang, B.; Niu, X. Soil Erosion Reduction by Grain for Green Project in Desertification Areas of Northern China. Forests, 2020, 11, 473.
That is, according to the distributed calculation method, the total evaluation object is decomposed into homogeneous ecological assessment units, and then the evaluation results are obtained by synthesizing them.
- This term has been published in Wang, B.; Niu, X.; Wei, W.J. National Forest Ecosystem Inventory System of China:Methodology and Applications. Forests 2020, 11, 732.
This method evaluated the ecological benefits associated with the GGP on five different levelss, each at a smaller scale of detail, i.e.: Level 1 - CPSR (14); Level 2 - administrative region (689); Level 3 - restoration method (3); Level 4 - forest type (3); Level 5 - dominant species. Ultimately 35,828 relatively homogeneous assessment units were determined.
- The data came from the survey of social and economic benefits of the GGP conducted by National Forestry and Grassland Administration and “Monitoring of Social and Economic Benefits of National Forestry Key Projects. System in other articles(Wu, X.; Wang, S.; Fu, B.; Feng, X.; Chen, Y. Socio-ecological changes on the Loess Plateau of China after Grain to Green Program. Total Environ. 2019, 678, 565-573. ) such as the output of grain, gross output value of agriculture, forestry, animal husbandry and fishery (AFAHF), number of rural employees are all include in the survey.
- We have enriched the content.
As a Nature based Solutions, the Grain for Green Project produced multiple benefit in the contiguous poverty-stricken regions and will help realize the sustainable development goals in China. The Grain for Green Project in the contiguous poverty-stricken regions has achieved “win-win” gains of restoring the environment and promoting socioeconomic development. The ecological benefit of the Grain for Green Projec in he contiguous poverty-stricken regions was 5601.21×108 Yuan/a in 2017, the proportion of each function being 27.1%(water conservation), 21.1%(biodiversity conservation), 18.4%(purification of atmospheric environment), 13.1%(soil conservation), 12.9%(carbon sequestration and oxygen release), 5.39%(forest protection), 1.6%(nutrient accumulation). Among the 14 areaa of the project, the Qinba Mountain area had the largest ecological benefit accounting for 19.3% of the total, followed by the Wuling Mountain area, the Liupan Mountain area, the Yunnan-Guizhou-Guangxi rock desert area, and the Wumeng Mountain area. In the contiguous poverty-stricken regions, 31.3% of the recorded poor households participated in the GGP, the project changed the production methods of farmers, the forestry industry was promoted and the income of farmers was increased, with an average increase of 0.51×104 Yuan/a per household. Further research should focus on interaction effects and measures to improve social-economic-ecological benefits.
Reviewer 3 Report
My comments are appended in the manuscript.
Moreover, results and discussion/ Conclusions needs further strengthening.

Author Response
Response to the Review Comments
We would like to thank you for your careful reading, helpful comments, and constructive suggestions, which has significantly improved the presentation of our manuscript.
We have carefully considered all comments from the reviewers and revised our manuscript accordingly. The manuscript has also been double-checked, and the typos and grammar errors we found have been corrected. In the following section, we summarize our responses to each comment from the reviewers. We believe that our responses have well addressed all concerns from the reviewers. We hope our revised manuscript can be accepted for publication.
- We have rewrite the abstract as following.
The Grain for Green Project(GGP) is a NbS which aims to address the challenge of ecological restoration while providing providing livelihood security to farmers in a poverty dominated region. Evaluating the success of such a project can prove difficult
- Line36 delete”objectively”
- Line43 delete “certain”
- line46 removed duplicate statements
- Line 49 change to minimal
- Line 57 This part is an objective description of the regional overview of the project implementation
- Line66 removed duplicate statements
- Line74-77 We think this sentence is suitable to the end of this paragraph. But we remove the following sentence to the next paragraph.
- Line 81 We made some changes. As a type of NbS, there is a lack of systematic review regarding the assessment and quantification of the ecological and economic effects of GGP and how it can solve social problem.
- We have corrected the Waste land Afforestation
- The figure quality had improved
- Line179 We have corrected the data
- We have transformed the table to figure
- Line244 we have removed duplicate statements
- We have rewrite the conclusion
As a Nature based Solutions, the Grain for Green Project produced multiple benefit in the contiguous poverty-stricken regions and will help realize the sustainable development goals in China. The Grain for Green Project in the contiguous poverty-stricken regions has achieved “win-win” gains of restoring the environment and promoting socioeconomic development. The ecological benefit of the Grain for Green Projec in he contiguous poverty-stricken regions was 5601.21×108 Yuan/a in 2017, the proportion of each function being 27.1%(water conservation), 21.1%(biodiversity conservation), 18.4%(purification of atmospheric environment), 13.1%(soil conservation), 12.9%(carbon sequestration and oxygen release), 5.39%(forest protection), 1.6%(nutrient accumulation). Among the 14 areaa of the project, the Qinba Mountain area had the largest ecological benefit accounting for 19.3% of the total, followed by the Wuling Mountain area, the Liupan Mountain area, the Yunnan-Guizhou-Guangxi rock desert area, and the Wumeng Mountain area. In the contiguous poverty-stricken regions, 31.3% of the recorded poor households participated in the GGP, the project changed the production methods of farmers, the forestry industry was promoted and the income of farmers was increased, with an average increase of 0.51×104 Yuan/a per household. Further research should focus on interaction effects and measures to improve social-economic-ecological benefits.
Round 2
Reviewer 1 Report
This version is much improved. However, the major issues raised in the previous reference report are not well addressed enough.
Author Response
Thank you very much for your recommendation.We have tried our best to revise the manuscript according to your kind and construction comments and suggestions. We sincerely hope that this revised manuscript has addressed all your comments and suggestions.
- The ecosystem services mentioned in this study was the flow ,which is produced by forests in the state of 2017. The social-economic benefits were compared with those who are not participated in the GGP, and we evaluated the changes of these farmers and households after participating in the
- The indicator was selected by the State Forestry and Grassland Administrationand also partly used in similar studies such as(Wu, X.; Wang, S.; Fu, B.; Feng, X.; Chen, Y. Socio-ecological changes on the Loess Plateau of China after Grain to Green Program. Total Environ. 2019, 678, 565-573.). There are 659 counties in the CPSR which is also the main survey objects for the monitoring of social-economic benefits, and the object of investigation was nearly 1/6 of them. What’s more, questionnaire survey conducted in 25 provinces in China covering over 8000 farmer households as the added data. Last but not least, the g data from the departments of statistics, agriculture, water conservancy, land, environmental protection, meteorology and other similar departments was used to assure the reality of data. We have added more detail information on line 172-178.
Reviewer 2 Report
Accept in present form
Author Response
Thank you for your suggestions and acknowledgment of this manuscript.